# Bryophytes Collection of the University of Brasilia Herbarium, Brazil

Mel C. Camelo *, Allan L. A. Faria, Daniela Cemin, Paulo E. A. S. Câmara and Micheline Carvalho-Silva *

Departamento de Botânica, Instituto de Ciências Biológicas, Universidade de Brasília, Brasília 70910-900, Brazil; allanlaid@gmail.com (A.L.A.F.); danicemin01@gmail.com (D.C.); paducamara@gmail.com (P.E.A.S.C.)
* Correspondence: melbiologia2010@hotmail.com (M.C.C.); silvamicheline@gmail.com (M.C.-S.)

**Abstract:** The UB Herbarium, located in the Department of Botany at the University of Brasilia (Brasilia, Brazil), was established in 1963. It is the third-largest herbarium in Brazil, housing approximately 277,000 samples. This study presents a quantitative description of the bryophytes collection at the UB Herbarium, which is the second-largest bryophytes collection in Brazil. It contains 31,099 samples, including specimens from all continents and 79 countries, with a focus on specimens from Brazil, Papua New Guinea, Malaysia, the United States, Chile, Indonesia, South Africa, Ireland, Argentina, and Sweden, as well as various islands and archipelagos. The collection has grown significantly since its creation in 1963, when it initially held 869 specimens; it now contains 31,099 specimens, which is a 59.3% increase. The herbarium holds 95 types of bryophytes. These results were gathered from consultations in the UB Herbarium online database and compiled into an Excel spreadsheet. These findings highlight the importance of our collection, making it a valuable resource for students and researchers interested in exploring and studying a diverse array of specimens.

**Keywords:** biological collections; herbaria; historical collections; Index herbariorum; taxonomy; types

## 1. Introduction

Biological collections are a vital aspect of our efforts to comprehend and safeguard life on Earth. They are meticulously curated and housed by natural history museums, research institutes, and universities, covering a wide variety of organisms from invertebrates and vertebrates to plants. These collections are an integral part of the ex situ conservation strategy and play a crucial role in global research infrastructure, supporting conservation efforts, scientific discoveries, and technological innovations [1].

Among these collections, herbaria stand out as repositories for preserved plant specimens, encompassing a broad spectrum of organisms like fungi, lichens, and algae. Their significance lies in their role as invaluable reservoirs of data documenting Earth's plant diversity, playing a fundamental role in the study of genomics and the impacts of climate change. Herbaria also house nomenclatural types, making them essential for studies focusing on threatened species and for documenting existing taxa [2].

Accessing historical herbarium specimens grants researchers a unique vantage point into the past, enabling investigations into shifts in plant distributions, phenological patterns, and overall biodiversity over time. This historical perspective is crucial for understanding the impacts of environmental changes, human interventions, and other factors on plant life and ecosystems [3].

The scale of these collections is staggering, with more than 396 million specimens preserved across over 3400 active herbaria globally, forming the foundational documentation for formally described plant species [3,4] The largest herbaria, such as P—Paris, NY—New York, K—Kew, MO—Missouri, and LE—St. Petersburg, collectively house approximately 36 million specimens. To enhance accessibility, many herbaria are actively digitizing their

collections and making images available online. While complete digitization remains a work in progress, there are already about 400 million digitized specimens, with this number steadily increasing [5–7].

As of April 2024, the Global Biodiversity Information Facility has documented over 111 million preserved specimen records for plants alone, highlighting the global scope and significance of these collections [8].

In Brazil, the importance of herbaria is reflected in the presence of 216 active herbaria as of 2018, with 162 registered in the Index Herbariorum. These herbaria collectively house over 6.7 million samples, including more than 39,000 type samples. Most Brazilian herbaria (67%) are in universities [9]. The top five Brazilian herbaria in terms of specimen numbers are RB (Jardim Botânico do Rio de Janeiro) with 824,457 specimens, MBM (Museu Botânico de Curitiba) with 332,759 specimens, UB (Universidade de Brasília) with 276,620 specimens, INPA (Instituto Nacional de Pesquisas da Amazônia) with 269,445 specimens, and HUEFS (Universidade Estadual de Feira de Santana) with 268,574 specimens [10].

Furthermore, the Brazilian Central region is home to 21 herbaria, each contributing significantly to botanical knowledge and research endeavors [9,10]. They are an essential repository of biodiversity data, supporting scientific research in the environmental field, especially botany and ecology. However, despite their scientific significance, it is surprising how little institutional recognition their curators receive. This may be due to an academic perception that biological collections are linked to traditional research rather than a hub for cutting-edge science [11]. Therefore, long-term policies and institutional recognition are essential to consolidate the advances made in the areas of taxonomy and biological collections in Brazil [12].

The importance of herbarium studies goes beyond recent specimens to include historical collections. This broad approach is vital in various domains, from traditional herbarium studies to cutting-edge research in climate change, biodiversity conservation, phenology, biological invasions, DNA banking, phylogenomics, and beyond [13–20].

In essence, herbaria and biological collections are not just repositories of specimens but reservoirs of invaluable information essential for understanding our planet's past, present, and future. They are integral to scientific progress, environmental stewardship, and the conservation of Earth's rich biological heritage.

The Herbarium of the University of Brasilia (international code: UB) was established in 1963 by Dr. João Murça Pires (1917–1994) and Dr. Graziela Maciel Barroso (1912–2023). They were the first taxonomists responsible for the UB Herbarium. Notable botanists, such as Ezechias Paulo Heringer (1905–1987), also contributed to the compilation and cataloging of botanical specimens from the Brazilian Central region [21].

Over time, the institution has grown into a significant reference center for taxonomic and floristics studies in the Brazilian Central region. Today, it stands as the largest Brazilian central region herbarium and holds global recognition. It contains specimens from various Brazilian vegetation types, as well as collections from different parts of Latin America and from around the world [22].

However, it is estimated that herbaria contain over 35,000 undescribed and newly discovered species [23]. These specimens, representing new species, remain undetected and undescribed because they may be inaccessible, because their information is incomplete, or due to a lack of required expertise for analysis. These new species are then unnoticed, misplaced, or treated as unidentified material. Thousands and thousands of sheets are still not identified at the species level, while numerous sheets must be reviewed and updated following more recent taxonomic knowledge [23].

The accurate species recognition underpins our knowledge of global biodiversity [23–25]. In recent years, the lack of taxonomic activity has led to increased political and scientific calls [25] to invest in the science of taxonomy, which is essential for what we know about species-level diversity. The assumptions behind these demands are that increased resources would necessarily lead to increased taxonomic productivity and accuracy. Given finite resources, scientifically sound criteria regarding where funds should most usefully be

targeted must be used to determine priorities for taxonomic research. It is therefore surprising that the processes of collecting, recognizing, and describing species are poorly understood and only rarely discussed [26–30] and that there is little research focused on the processes resulting in the recognition of new species. For many groups of organisms, so little information is available about them that measuring any aspect of the discovery process suffers from a lack of data.

Furthermore, the increasing global anthropogenic-driven pressure affects organisms worldwide and leads to unprecedented biodiversity loss. Out of ca. 1800 bryophyte species occurring in Europe, 34% are assessed as threatened or near-threatened [30].

The UB Herbarium houses a valuable collection of bryophytes, including liverworts (Division Marchantiophyta), mosses (Division Bryophyta), and hornworts (Division Anthocerotophyta). Although there are no specific records of the date of its creation, the bryological collection originated in the 1960s with a few specimens primarily collected by undergraduate Biology students at the University of Brasilia. The collection is the culmination of decades of collection and research efforts, contributing to the understanding of a botanical group that has been relatively underexplored. In 1979, we received a new crytogamist specialist Dr. Lauro Xavier Filho at UnB. Dr. Lauro Xavier remained at UnB until 1982 when he was succeeded by the phycologist Dr. Pedro Américo Cabral Senna, who played a significant role in the collection's growth [22].

The bryophyte collection at UB Herbarium saw substantial growth starting from the late 1960s and early 1970s, mainly due to collections by H.S. Irwin and contributions from Daniel Moreira Vital. Vital's expeditions, initiated by Dr. Pedro Américo in 1985, covered various locations in the Distrito Federal, significantly adding to the collection's diversity. When Dr. Pedro Américo departed from UnB in 1992, he left behind a collection of 1100 bryophyte specimens primarily from the Distrito Federal, collected by Irwin, Vital, and their students.

In 2002, Paulo E.A.S. Câmara completed the first master's dissertation focusing on the Brazilian Central region bryophyte flora, further enhancing the collection. In 2009, the hiring of Dr. Paulo Câmara as the first bryophyte specialist marked a new phase, with the collection expanding to around 1600 specimens—a growth of about 500 specimens in 17 years. His expertise facilitated the identification of numerous plant species, and his collaborations with master's and doctoral students enabled extensive collections throughout Brazil, significantly enhancing the herbarium's holdings [22].

The culmination of these efforts was the publication of "Flora do Distrito Federal: Briófitas" in 2017, which was facilitated by the UB Herbarium's bryophyte collection [31]. This collection became the most extensive in the region, containing many taxa found exclusively within it. Overall, the strategic contributions of collectors, specialists like Dr. Paulo Câmara, and collaborative research efforts significantly expanded and enriched the bryphyte collection at UB Herbarium, making it a vital resource for studying the flora of the Distrito Federal and beyond.

Another significant factor was the receipt of duplicates from Dr. Paulo Câmara's collections in Southeast Asia, a result of his doctoral work at the Missouri Botanical Garden. This expanded the regional nature of the bryophyte collection, which was further enhanced by the exchange of specimens from various herbaria, including BM, CAS, L, MO, NY, PRE, S,W, and national herbaria such as FLOR, HPAN, RB, SP, and TANG. The exchange program has contributed to UB's collection by providing rare plants, including specimens collected in the 19th and 20th centuries by renowned collectors. Notably, the herbarium has received contributions from pioneers in polar research, such as Robert Peary [22].

Furthermore, in recent years, herbaria worldwide have been publishing their bryophyte collection database, showing the number of specimens, families, genera, and types represented. This transparency opens up opportunities for further botanical research and collaboration [30–34].

The presence of numerous taxa not found in Brazil is of paramount importance for research and the training of future taxonomists. The bryophyte collection at UB plays a

vital role in supporting studies and facilitating the development of taxonomic expertise, for example, many studies of flora, new species, and new records of bryophytes have come out over these years [35–48].

This work aims to present the dataset of UB Herbarium's bryophyte collection, with a focus on its representation of both Brazilian and global flora. The collection includes specimens from every continent, a variety of countries, and numerous islands. The herbarium's work is ongoing, with regular field collections and new specimens arriving each year, leading to continuous updates.

## 2. Materials and Methods

The bryophytes collection was organized into three main taxonomic groups: mosses (Bryophyta), liverworts (Marchantiophyta), and hornworts (Anthocerotophyta) and mounted in exsiccates, preserved at low temperatures, in a controlled environment. The names of divisions and families are as described in Part 3 of A. Engler's Syllabus of plant families, Bryophytes, and seedless vascular plants [48]. The collection has acquired a new space and has been physically separated from the collections of other plants due to the increasing need for physical space (Figure 1).

After field collection, delicate bryophyte structures such as the thallus, sporophytes, and antheridia of liverworts and hornworts are preserved using FAA, Transeau solution, or 1% formalin.

Materials used for herborization include standardized envelopes (12.8 × 9.5 cm) made of white bond paper (28 × 21.5 cm). To dry bryophytes, the material is spread out between layers of newspaper and gently pressed. After drying at room temperature or in an oven (in humid tropical regions), the material is placed in paper bags and then stored in standardized envelopes made of white bond paper. These envelopes are labeled and kept in drawers [49,50].

Label data should include the name of the species (if known), the author of the scientific name; the altitude, habitat, substrate, date (including the month) and location (country, state, county, distance to nearest town) of collection; the GPS coordinates; the name of the collector; the collection number and determiner (the name of the person identifying or verifying identification). Additional information may include the name of the associated species, color, plant height, abundance, or other information not evident from the pressed specimen. For liverworts, descriptions of the oil bodies should be included as these will disappear upon drying. The label usually also includes the name of the herbarium and the accession number for that herbarium. The herbarium name ensures that any loans are returned to the rightful owner [50].

The types of specimens are separated in colored folders traditionally used for tracheophytes to indicate special collections. Red is standard for type specimens, whereas blue or other colors may be used to indicate a particular geographic area. The same system can be used if bryophytes are stored on herbarium sheets and provides one of the arguments in favor of this method. A red felt pen run across the top of a packet will serve the same purpose, or a red herbarium folder can be cut to fit around the packet.

To preserve the material, as bryophytes thrive in low temperatures, it is not surprising that cooling them during drying can improve the sample quality during air drying. Gametophytes and sporophytes remain in the best condition when fresh samples, still in their paper bags, are placed in a freezer for 15 days at 7 °C, 37% relative humidity. The method discourages fungal growth and retains colors, leaf details, stem structure, and various types of bryophytes, including leafy liverworts, thalloid liverworts, and hornworts. The low temperature slows dehydration through a more natural approach [50].

We analyzed the total number of deposited specimens, the total number of specimen collections for each continent, the top ten countries of origin, the top ten collectors and determin ers, and the most represented taxa among mosses and liverworts. The life forms were sur veyed using a Microsoft Excel spreadsheet loaded on the Jardim Botânico do Rio de Janeiro (RB) website and UB Herbarium was selected (http://ub.jbrj.gov.br/

v2/consulta.php, accessed on 13 March 2024), and data were also gathered from the Specieslink website (https://specieslink.net/search/, accessed on 13 March 2024). The survey encompasses the entire period from the date the herbarium first opened to the present day (2024). The tables and graphics were created by a Microsoft Excel spreadsheet loaded on the Jardim Botânico do Rio de Janeiro (RB) website and UB Herbarium was selected (http://ub.jbrj.gov.br/v2/consulta.php, accessed on 13 March 2024).

Maps were generated to illustrate the global distribution of deposited specimens, including their representation in Brazil and its phytogeographical domains. The most significant areas are highlighted. The maps were created using QGIS 3.22.16-Białowieża (https://qgis.org, accessed on 13 March 2024), and shapefiles (vectors) for the world, Brazil, and Brazil's phytogeographical domains according to IBGE 2012 were used [51]. The heat map (Kernel Density Estimation) was employed for Brazil maps in the QGIS software [51] to visualize regions with a higher density of records. The parameters used for Brazil include a radius of 100,000 m, 4480 lines, 4381 columns, pixel size x and y: 1 thousand, Kernel-shape Quartic, SRC: ESG:5880—SIRGAS 2000/Brazil Polyconic, rendering type: single-band false-color, Band 1 (Gray), and cumulative cut count [51]. Phytogeographic domains are based on IBGE and a review of plant biogeographic studies in Brazil [52,53].

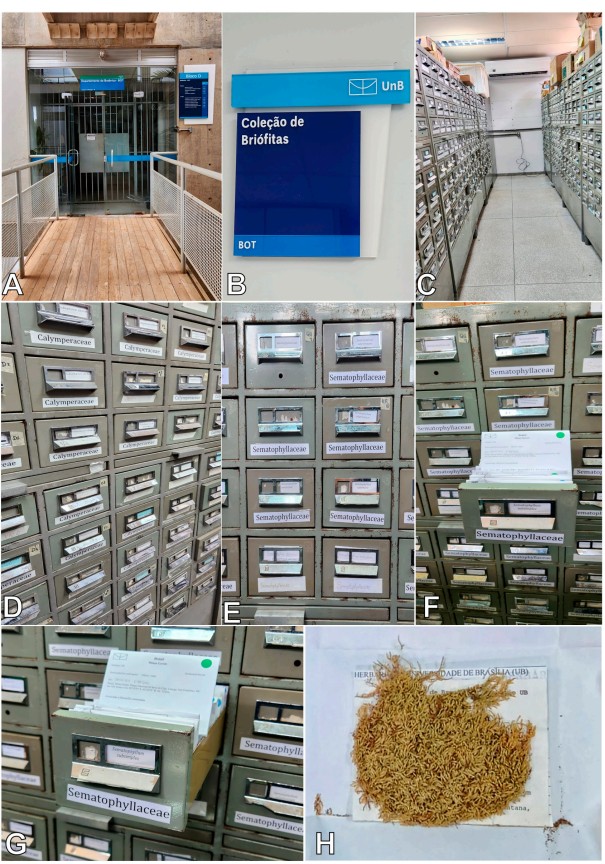

**Figure 1.** Bryophyte collection of the UB Herbarium. (**A**) Botany Department at the University of Brasília. (**B**) Plate indicating the bryophyte collection. (**C**) Cabinets. (**D**) Cabinets with small drawers adapted for bryophyte species and details of Calymperaceae cabinets. (**E**) Sematophyllaceae cabinets. (**F**) Details of Sematophyllaceae cabinets showing one letter. (**G**) Letter of Sematophyllaceae indicated with a green dot (collected in Brazil). (**H**) One specimen of Sematophyllaceae.

## 3. Results

### 3.1. Geographical Coverage

The bryophyte collection at the UB Herbarium is currently represented by 31,099 specimens, of which 5000 are undetermined (Figure 2). The collection includes specimens from

every continent: South America (21,673 specimens), Antarctica (8149 specimens), Asia (2593 specimens), Oceania (2075 specimens), Europe (1370 specimens), North America (1092 specimens), and Africa (902 specimens) (Figure 2). Seventy-nine countries are represented with collections. The countries that are most frequently collected from are Brazil (17,680 specimens), Papua New Guinea (1030 specimens), Malaysia (791 specimens), the U.S.A. (710 specimens), Chile (526 specimens), Indonesia (506 specimens), South Africa (501 specimens), Ireland (466 specimens), Argentina (430 specimens), Sweden (331 specimens), Bolivia (108 specimens), Japan (107 specimens), Mexico (86 specimens), Spain (50 specimens), Peru (46 specimens), Netherlands (40 specimens), Finland (37 specimens), Australia (36 specimens), Sao Tome e Principe (30 specimens), and Thailand (26 specimens). Other important collections have been obtained from islands and archipelagos, with 11,777 records, and most of these are from Antarctic Islands (4940 specimens), Falklands (Malvinas) (754 specimens), South Georgia (696 specimens), Trindade Island (601 specimens), and Fernando de Noronha (558 specimens).

The top ten collectors of bryophytes are Câmara, P.E.A.S. with 3879; Soares, A.E.R. with 2079; Valente, D.V. with 1636; Faria, A.L.A. with 1204; Gama, R. with 934; Carvalho-Silva, M. with 840; Souza, R.V. with 827; Dantas, T.S. with 687; Vital, D.M. with 526; and Cunha, M.J. with 338. The top 10 determiners of the specimens are Câmara, P.E.A.S. with 3336; Peralta, D.F. with 2074; Souza, R.V. with 1342; Faria, A.L.A. with 820; Soares, A.E.R. with 740; Evangelista, M. with 483; Cunha, M.J. with 389; Souza, R.V. with 149, Carvalho-Silva, M. with 109; and Steere, W.R. with 80 specimen determinations.

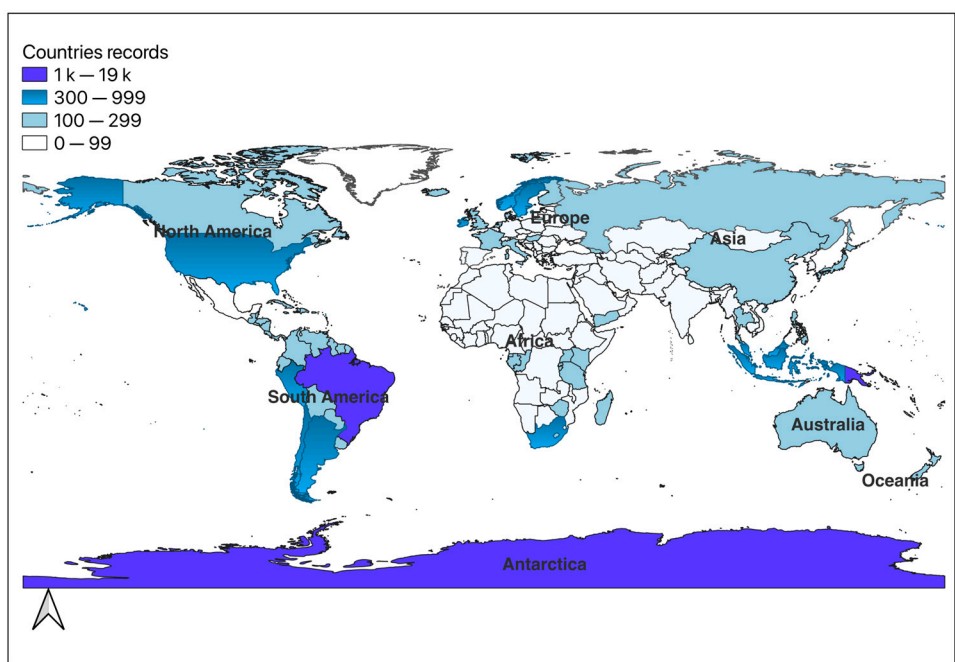

**Figure 2.** Records of bryophytes in countries collected and deposited at UB Herbarium.

In Brazil, the main regions of collection are the Midwest and Southeast, including the Distrito Federal, Goiás, and Mato Grosso, followed by São Paulo, Minas Gerais, and Rio de Janeiro. The phytogeographic domains in Brazil from which the most specimens have been collected are Cerrado, the Atlantic Forest, and the Amazon (Figure 3). However, the herbarium has the best collections for the Brazilian islands of Fernando de Noronha and Trindade Island (Figures 3 and 4).

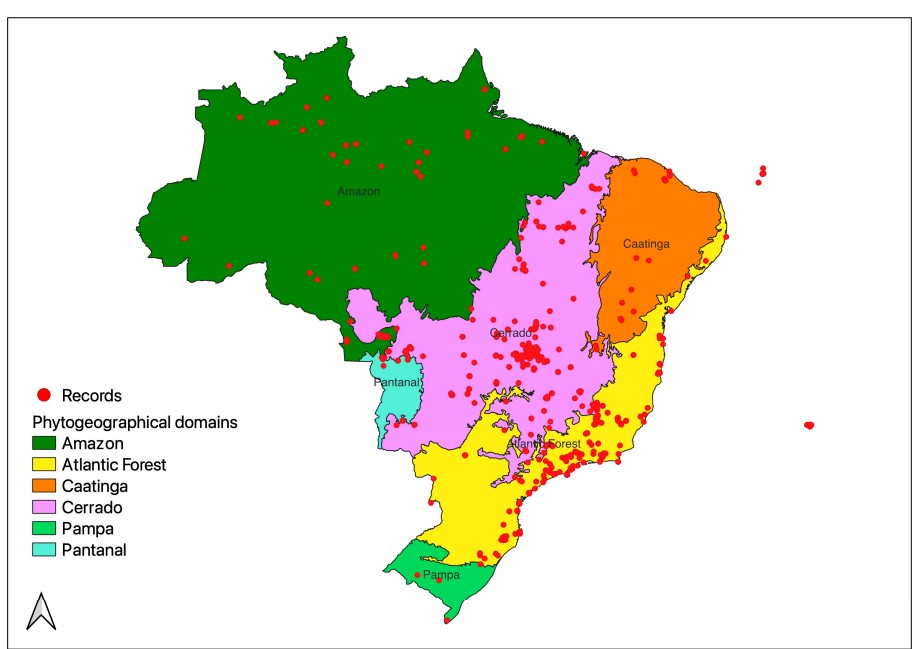

**Figure 3.** Bryophyte records from phytogeographical domains of Brazil deposited at the UB Herbarium.

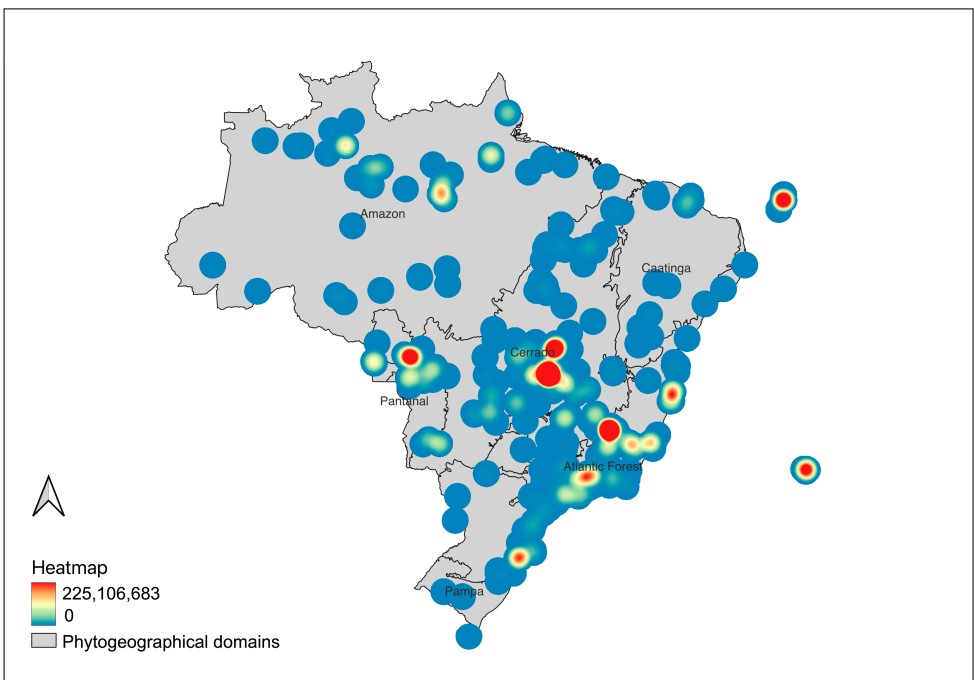

**Figure 4.** Heatmap (Kernel density) of bryophyte records from the phytogeographical domains in Brazil.

### 3.2. Temporal Coverage

The exchange program has contributed to UB's bryophyte collection by providing specimens collected in the 19th and 20th centuries by renowned collectors. Notably, the herbarium has received contributions from pioneers in polar research, such as Robert Peary. The collection includes specimens collected from 1840 to the present day (Figure 5). The oldest specimens correspond to *Bartramia ithyphylla* Brid. collected by Jan Wttewaall s.n in the Netherlands, Gelderland, in 4/1840.

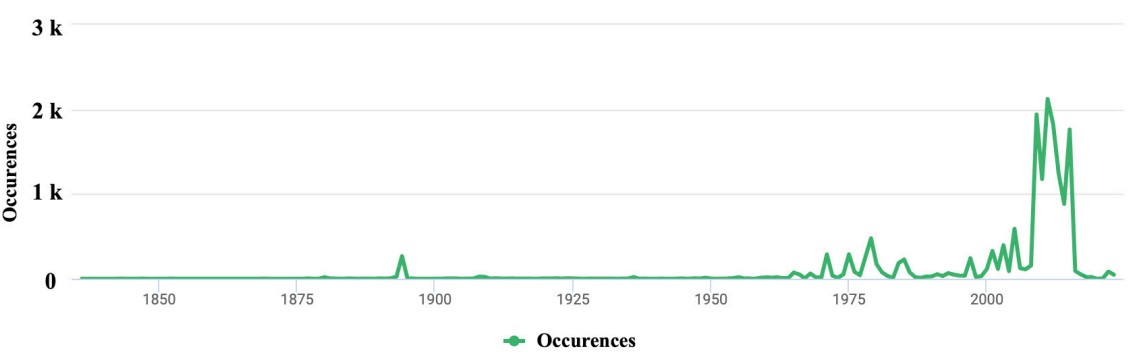

**Figure 5.** Growth of the UB dataset (number of specimen entries) per year.

Based on the annual growth of the UB dataset (number of specimen entries), the collection contained 200 specimens in 1880, 500 specimens in 1978, and approximately 2000 entries by 2010 (see Figure 5). At its creation in 1963, the collection held 869 specimens. By 2014, this number had increased to 19,518 specimens, marking a 34.78% growth. Comparing 2014 to 2024, the collection has grown to 31,099 specimens, reflecting an increase of 59.3% (see Figure 6).

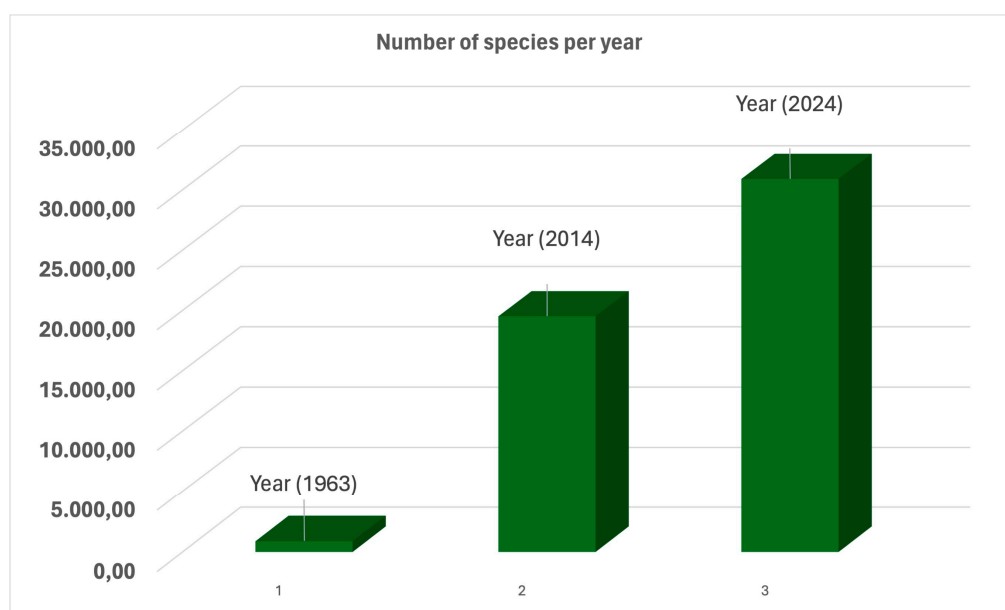

**Figure 6.** Chronological distribution of bryophytes specimens in UB herbarium: Number of specimens × year of collection. Captions: Year 1: 1963 (date of creation), Year 2: 2014, Year 3: 2024.

### 3.3. Taxonomical Coverage

Most of the records pertain to mosses (93%), followed by liverworts (6%), and 1% pertain to hornworts. There are 27,353 specimens of the Division Bryophyta Schimp. Musci, and the most representative families are Bryaceae Schwägr., comprising 21% (3961); Sematophyllaceae Broth, comprising 11% (3742); Pottiaceae Schimp., comprising 11% (1973); Polytrichaceae Schwägr., comprising 10% (1733); Leucobryaceae Schimp., comprising 9% (1648); Hypnaceae Schimp., comprising 8% (1325); Pylaisiadelphaceae Goffinet & W.R.Buck, comprising 7% (1160); Calymperaceae Kindb., comprising 7% (1215); Bartramiaceae Schwägr., comprising 5% (945); Fissidentaceae Schimp., comprising 5% (859); and Dicranaceae Schimp., comprising 5% (796), while other families make up 1%.

Subsequently, there are 1924 specimens of the Division Marchantiophyta Stotler & Crand.-Stotler, with the most representative families being Lejeuneaceae Cavers, comprising 44% of the records (1144); Plagiochilaceae (Joerg.) K.Müll., comprising 14% (364);

Frullaniaceae Raddi, comprising 12% (304); Metzgeriaceae Raddi, comprising 7% (187); Lepidoziaceae Limpr., comprising 7% (177); Radulaceae K. Müll., comprising 3% (85); Lophocoleaceae De Not., comprising 4% (100); Pallaviciniaceae Mig., comprising 3% (83); Marchantiaceae (Bisch.) Lindl., comprising 3% (78); Aneuraceae H.Klinggr., comprising 2% (68); and other families comprising 1%.

Lastly, there are 44 specimens of the Division Anthocerotophyta Rothm. ex Stotler & Crand.-Stotl., with the most representative families being Notothyladaceae Grolle, comprising 64% of the records (29); Anthocerotaceae Dumort., comprising 27% (12); Dendrocerotaceae J. Haseg., comprising 7% (3); and other families comprising 2%.

Notably, the collection includes 95 types of materials: 7 holotypes, 27 isotypes, 10 syntypes, 1 isosintype, and 1 paratype, while 48 types are not specified. Most of the specimens belong to the families Hypnaceae, Leskeaceae Schimp., Mniaceae Schwägr., Pylaisiadelphaceae, Sematophyllaceae, Sphagnaceae Dumort., and Thuidiaceae Schimp. (Supplementary Table S1).

The type materials were collected by renowned naturalist and botanists like Friedrich Wilhelm Heinrich Alexander von Humboldt, Richard Spruce, Howard Samuel Irwin, Auguste François Marie Glaziou, James Bisset, Eduard Friedrich Poeppig, Sir Ferdinand Jakob Heinrich von Mueller, Joseph Dalton Hooker, Ernest Heinrich Ule, Cyrus Guernsey Pringle, Alvan Wentworth Chapman, William Russel Buck, and others (Supplementary Table S1).

## 4. Discussion

The bryophyte collection at UB Herbarium is the second largest in Brazil, exceeded only by the SP Herbarium, which comprises around 31,099 specimens from 79 countries [10]. UB's collection presents samples collected from all continents, including Antarctica and representatives from little-known oceanic islands with few collections due to lack of accessibility, such as Trindade, Falklands (Malvinas), and South Georgia, and covers all the Brazilian phytogeographical domains. UB's collection presents the largest collection of Antarctic plants in comparison with other known Brazilian herbaria and the largest of Antarctic plants in Latin America, comprising approximately 8149 specimens [22].

The predominance of mosses deposited in the herbarium was expected, considering that this division is the most abundant in Brazil, with 896 species, and globally there are 12,900 species [54,55]. The high diversity of mosses can be explained by their morphology, as the structure of their gametophyte and the different life forms within this group allow them to thrive in dry, exposed, or high-altitude environments [56–58].

The mosses' abundant families were Sematophyllaceae, Bryaceae, and Pottiaceae, listed as frequent in tropical forests [59]. Additionally, Bryaceae is one of the families with the broadest richness and distribution, encompassing approximately 660 cosmopolitan species [59]. Regarding the division Marchantiophyta, Lejeuneaceae was the richest family, possibly due to the affinity of its representatives with common substrates in forested areas, such as branches, trunks of living or decomposing trees, rocks, soils, and living leaves [55].

Concerning the division Anthocerotophyta, hornworts are the least diverse of the bryophytes, comprising around 300 species worldwide and 23 taxa confirmed occurrences in Brazil. This lower species incidence was expected [56,60].

## 5. Conclusions

An analysis of the bryophyte samples in the UB Herbarium underscores the importance of our collection. With specimens from all continents and across all Brazilian phytogeographical domains, we can conclude that our herbarium serves as a valuable resource for students and researchers seeking to explore and study our diverse range of bryophyte specimens.

We have a few samples deposited in this collection from the African, North American, and European continents, as well as the North and Northeast regions of Brazil. This scarcity of bryophyte samples is mainly due to the lack of relationship that researchers at the UB Herbarium have with institutions on these continents, there are currently no research

projects with bryophytes that involves such regions in addition to the UB Herbarium. This also reduces the exchanges of samples made from one herbarium to other herbaria.

Regarding phytogeographic domains in Brazil, the low sampling of bryophytes in the Amazon, Caatinga, and Pampa domains in the UB Herbarium is mainly related to the lack of professionals in the field who need to collect bryophytes for taxonomic and ecological studies. The North and Northeast regions, in addition to being the largest regions in Brazil, are areas that are underexplored due to the number of bryophyte researchers in these regions, as several states still lack bryophyte professionals in their research institutions.

The significance of specimens deposited in our herbarium collected on islands and archipelagos is extremely important because many of the species are already categorized as under threat. This facet has not only proven to be vital for our herbarium but also serves as a distinguishing factor. Moreover, we recognize the importance of acquiring more specimens of hornworts, as this will enhance the richness of our collection and facilitate additional scientific studies.

**Supplementary Materials:** The following supporting information can be downloaded at: https://www.mdpi.com/article/10.3390/d16060342/s1, Table S1: Type specimens of UB Herbarium.

**Author Contributions:** M.C.C. contributed to conceptualization, data curation, formal analysis, investigation, methodology, software, writing—original draft preparation; A.L.A.F. to writing—original draft preparation; D.C. contributed to data curation, formal analysis, and methodology; P.E.A.S.C. and M.C.-S. were responsible for data curation, investigation, supervision, project administration, resources, validation, visualization and writing—original draft preparation. All authors have read and agreed to the published version of the manuscript.

**Funding:** This research has no funding and the APC was funded by Universidade de Brasilia, edital DPI/DPG/DCE n. 01/2024, formulary n. 11226785.

**Institutional Review Board Statement:** Not applicable.

**Data Availability Statement:** Data are contained within the article.

**Acknowledgments:** We would like to thank CAPES-BRASIL the postdoctoral fellowship granted to the first author under project no "88887.800986/2023-00". We are indebted to the curators and technicians of Herbaria UB for collecting many species from the central region and contributing to botanical identifications. We thank Simon J. Mayo (Kew Gardens) and Andressa Carolina Vieira dos Santos Corrêa de Oliveira (Flyover Ensino de Idiomas) for checking the language of this article. Collections from Antarctica were possible due to the support of the Brazilian Antarctic Program (PROANTAR) and the Secretaria da Comissão Interministerial para Recursos do Mar (SECIRM), who also facilitated collections on remote islands. A series of grants also made it possible to collect material under the PROTAX, PROTRINDADE and PNPD programs.

**Conflicts of Interest:** The authors declare no conflicts of interest.

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
