# Peer review of "Bryophytes Collection of the University of Brasilia Herbarium, Brazil"

_diversity, doi:10.3390/d16060342_

Round 1
Reviewer 1 Report (Previous Reviewer 5)
Comments and Suggestions for Authors
I believe that the authors have taken on board and responded to the comments and suggestions made. Therefore the manuscript has been improved and now deserves publication.
Author Response
Thank you for your considerations and review of the text.
Reviewer 2 Report (Previous Reviewer 4)
Comments and Suggestions for Authors
The manuscript has been improved in some aspects but still suffers from multiple issues. The first of them is a sometimes the unclear structure or logic of the text at some places (e.g., par 2 of Introduction - why the Brazilian Cerrado should be considered a specific unit for herbarium census; why "approaches addressing climate change etc." should be relevant to the number of active herbaria in Brazil....).
There remain multiple unclear points in the presented results (e.g., why the map in Fig. 2 shows >1000 specimens from USA while in the text you list 710...?). There is a rather funny understanding of 'ancient plants', seemingly meaning any specimen collected before ca. 1950. The description of herbarium growth is accomplished in way which I don't understand - you start with the year 1880 (when the herbarium did not exist), the percentages of increases are not clearly referred to some points. You should name the authors of types in the consistent way in all cases (sometimes you list full names, sometimes only initials or nothing is presented). I do not understand why life forms are presented for the specimens, moreover the thallose /foliose form in hepatic is not a life form, both foliose and thallose liverworts can appear in mats, tufts etc.
There is a very funny conclusion suggesting that the relative paucity of specimens from North America and Europe in herb. UB shows the necessity of a better exploration of these continents - do the authors believe that the exploration of any place on Earths necessarily results in the proportional representation of the specimens from such a place in herbarium UB? Similarly I can see no substantiation for the imperative at the sampling initiative in regions which are underrepresented in UB specimens (unless these regions are underrepresented in all Brazil's and world herbaria) - but such data have not been shown by the authors.
Comments on the Quality of English LanguageThe English in the manuscript is understandable, but often clumsy. In some cases, it appears that the authors probably want to convey something else than what the text sounds like.
Author Response
We made the necessary changes and attached the text file with the authors' comments.

Reviewer 3 Report (Previous Reviewer 3)
Comments and Suggestions for Authors
After the author revised all the comments in this manuscript, I think it is suitable to published on the journal. The bryophyte specimens in the UB Herbarium are important for researchers to do further research, such as the diversity and differentiation of bryophytes all around the world.
Author Response
Thank you for your considerations and review of the text.
Round 2
Reviewer 2 Report (Previous Reviewer 4)
Comments and Suggestions for Authors
While again seeing some improvements in the specific points raised previously, the overall quality of presentation and soundness of presented data has not improved dramatically. I can again raise multiple points which I can see as unclear or non-relevant, e.g.:
Introduction: par1 and 2 tells largely the same
Par3 - documentation of taxa that have become extinct is a minor part, most important info is on the extant species
At p2, li61 you still retain „Brazilian Cerrado“ – if you use this in the same sense as „Brazilian Central region, you should use it consistently
P7li227: You still retain the term „ancient plants“, contrary to the cover letter, specifying them (in part) as „specimens collected in the 19th and 20th century by renowned collectors“. I still believe the usage of term „ancient plant“ is inappropriate in this understanding.
P7li233-7: What do you mean by „In 1963, the date of creation we had 869 specimens an increase 34,78%.“? How can you have 1/3 increase upon the date of creation? Why do you compare the current state specifically with 2014?
In conclusions, I would mainly expect that you will summarize the strong points of UB herbarium, rather than admitting (although correctly of course) that you are lacking a good representation of European and other exotic specimens (which is rather logical and expected). The paragraph on specimens from islands does not appear to be directly related the preceding text. What is the meaning of the sentence „The significance of collecting specimens on islands cannot be overstated, as many of the species discovered are already under threat“? Similarly, I do not understand which „findings underscore the importance of our collection...“ – the sentence is not related to any particular finding of the presented text.
Comments on the Quality of English LanguageThe English of the text must be improved, clarity issues remain.
Author Response
Dear reviewer, we have included your suggestions in our article.
The language was checked by Dr. Simon Mayo (Kew Gardens).
While again seeing some improvements in the specific points raised previously, the overall quality of presentation and soundness of presented data has not improved dramatically. I can again raise multiple points which I can see as unclear or non-relevant, e.g.:
Introduction: par1 and 2 tells largely the same
Par3 - documentation of taxa that have become extinct is a minor part, most important info is on the extant species
R: We updated the introduction and include the information „Herbaria also house nomenclatural types, making them essential for studies focusing on threatened species and the documentation of extant taxa”.
At p2, li61 you still retain „Brazilian Cerrado“ – if you use this in the same sense as „Brazilian Central region, you should use it consistently
R: We choose to use Brazilian Central region.
P7li227: You still retain the term „ancient plants“, contrary to the cover letter, specifying them (in part) as „specimens collected in the 19th and 20th century by renowned collectors“. I still believe the usage of term „ancient plant“ is inappropriate in this understanding.
R: We adjust for only rare plants.
P7li233-7: What do you mean by „In 1963, the date of creation we had 869 specimens an increase 34,78%.“? How can you have 1/3 increase upon the date of creation? Why do you compare the current state specifically with 2014?
R: In 1963, the date of creation we had 869 specimens, and 2014, we had 19,518 specimens an increase of 34.78% a comparison of the 2014 year with now 2024, we have 31,099 specimens, an increase of 59,3 % samples in our herbarium.
We choose 2014 to have 10-year comparison with now.
In conclusions, I would mainly expect that you will summarize the strong points of UB herbarium, rather than admitting (although correctly of course) that you are lacking a good representation of European and other exotic specimens (which is rather logical and expected). The paragraph on specimens from islands does not appear to be directly related the preceding text. What is the meaning of the sentence „The significance of collecting specimens on islands cannot be overstated, as many of the species discovered are already under threat“? Similarly, I do not understand which „findings underscore the importance of our collection...“ – the sentence is not related to any particular finding of the presented text.
R: Collections on islands could be moderate because many of the species discovered there are already under the threat category.
We adjust for “The results present show it the importance of our collection, which specimens from all continents, making it an attractive resource for students and researchers interested in exploring and studying our diverse array of bryophytes specimens”.

Round 3
Reviewer 2 Report (Previous Reviewer 4)
Comments and Suggestions for Authors
The manuscript now has been revised substantially, resulting in overall improvement in all earlier criticised aspects.
I can see now only minor points in the Methods and Results section (such as the need for explaining that Cerrado in Fig. 3 and the paragraph above is the same as 'Central region' appearing elsewhere, or the absence of presentation of digitization efforts at UB [only the general information on the process in the world is mentioned at p. 2 but we would be more interested in the situation at UB]).
However, I still see the need for a bigger alteration in Discussion and Conclusions sections, namely:
- the first paragraph of Discussion would better fit into Introduction, as this is not a reaction to any result by the authors
- par2li620 - what do you mean by 'most diverse' - of what?
- li639 - what is your statement that UB is 'the largest of Antarctic plants in Latin America' based on? Are the numbers of Antarctic collections hosted at other Brazilian herbaria known? If yes, please refer to them
- li641 and following: of course the predominance of mosses is statistically unexpected (as the liverworts represent ca. 1/3 of bryophyte diversity - not sure about Brazil specifically but the rate will not be drastically different), so why liverworts are so underrepresented at UB?
- li654-5: 18 species or just 18 collections of hornworts in Brazil in total???
- li658-9: in this sentence, some words are superfluous, some words are missing, please revise the wording
- li671: low representation of other than Cerrado specimens is maybe simply because you are located in the Central (Cerrado) region, so logically the representation of the region is the best unless you have specific projects in other regions?
- li676: what do you mean by 'The significance of collecting specimens on Islands and archipelagos cannot be overstated'? Did you mean, on the opposite, that the good representation of island collections at UB must be emphasized?
- li679 - I would not attract the attention to the underrepresentation of the liverworts at UB in conclusions, it is logical that you acknowledge this deficiency
- last paragraph - it is unclear if obtaining the additional resources for the expansion of UB is just you wishful thinking or you plan specific actions to improve the situation. If you don't want to specify this, it might be better not to mention this (as the resources for improvement of the current state is perhaps a general characteristics of all herbaria worldwide)
Comments on the Quality of English LanguageWhile I acknowledge a significant improvement in the quality of English after the revision (large parts of the text are now completely correct), a minor revision remains necessary, as there are obviously parts of the text or alterations that followed the English revision which obviously were not revised by a native or proficient English user.
Author Response
The manuscript now has been revised substantially, resulting in overall improvement in all earlier criticised aspects.
I can see now only minor points in the Methods and Results section (such as the need for explaining that Cerrado in Fig. 3 and the paragraph above is the same as 'Central region' appearing elsewhere or the absence of presentation of digitization efforts at UB [only the general information on the process in the world is mentioned at p. 2 but we would be more interested in the situation at UB]).
R: Central region and Cerrado domain are two different things. In the text, Central region is formed by three states: Goiás, Mato Grosso and Mato Grosso do Sul, plus the Federal District, where Brasilia, the country's capital, is located.
In Fig. 3. is correct to include Cerrado domain, is richest savannah in diversity in the world and the second largest domain in the country (included the states of Goiás, Tocantins, Mato Grosso do Sul, southern Mato Grosso, western Minas Gerais, Federal District, western Bahia, southern Maranhão, western Piauí), therefore it will remain that way in the figure caption.
R: Our results presented are the specimens deposited in UB herbarium, with representatives from different countries. I believe that the reviewer did not understand the central purpose of our article. Both in the text and in figure 1. presented are specimens deposited in our herbarium.
However, I still see the need for a bigger alteration in Discussion and Conclusions sections, namely:
- the first paragraph of Discussion would better fit into Introduction, as this is not a reaction to any result by the authors
R: We reallocated into Introduction in the part of Brazilian herbaria.
par2li620 - what do you mean by 'most diverse' - of what?
R: This word is incorrect and we delete it.
- li639 - what is your statement that UB is 'the largest of Antarctic plants in Latin America' based on? Are the numbers of Antarctic collections hosted at other Brazilian herbaria known? If yes, please refer to them.
R: UB's collection presents the largest collection of Antarctic plants in comparison with other Brazilian herbaria known and the largest of Antarctic plants in Latin America, comprising approximately 8,149 specimens.
li641 and following: of course, the predominance of mosses is statistically unexpected (as the liverworts represent ca. 1/3 of bryophyte diversity - not sure about Brazil specifically but the rate will not be drastically different), so why liverworts are so underrepresented at UB?
R: Our herbarium have 1,924 specimens of Marchantiophyta, why is underrepresented for you?
We can justify this, because we do not have specialists of liverworts or few specialists are visiting the herbarium to determinating. In addition, to the low rate of liverwort collections that are deposited here.
- li654-5: 18 species or just 18 collections of hornworts in Brazil in total???
R: 23 taxa are confirmed in Brazil in total.
- li658-9: in this sentence, some words are superfluous, some words are missing, please revise the wording. O.K.
- li671: low representation of other than Cerrado specimens is maybe simply because you are located in the Central (Cerrado) region, so logically the representation of the region is the best unless you have specific projects in other regions?
R: Yes. Our herbarium has an impressive collection from the Cerrado domain because there was a project to catalog the plants of the Federal District in 1999 (located in the Central region) [29] and therefore, there are many more specimens from this phytogeographic domain than others [27]. University of Brasilia of Botany Department has more specialist in plants of Cerrado Domain.
- li676: what do you mean by 'The significance of collecting specimens on Islands and archipelagos cannot be overstated'? Did you mean, on the opposite, that the good representation of island collections at UB must be emphasized?
R: We adjust this sentence.
li679 - I would not attract the attention to the underrepresentation of the liverworts at UB in conclusions, it is logical that you acknowledge this deficiency.
R: O.K. we delete it.
last paragraph - it is unclear if obtaining the additional resources for the expansion of UB is just you wishful thinking or you plan specific actions to improve the situation. If you don't want to specify this, it might be better not to mention this (as the resources for improvement of the current state is perhaps a general characteristics of all herbaria worldwide).
R: O.K. we delete it.

This manuscript is a resubmission of an earlier submission. The following is a list of the peer review reports and author responses from that submission.
Round 1
Reviewer 1 Report
Comments and Suggestions for Authors
Review of "Bryophytes collection of the UB Herbarium, Brazil" by Mel C. Camelo et al.
In the introduction there are numerous repetitions of not only words or slogans but also entire sentences in the text. I have marked some of them for the authors, but it is now their responsibility to analyze the text very carefully to detect such repetitions.
As for the text itself, it is significant that there are a lot of words but little content, the authors make a very general argument on general topics, which sometimes has very little to do with the topic of the article or does not lead to anything specific.
Another equally important observation is that the authors cite relatively few articles, very few for an introduction. Even more so for an introduction to such a general topic. Unfortunately, what's worse, they quote one article in one entire paragraph, and sometimes even on one and half the page. I think this oversight is a serious mistake.
Another very important shortcoming is that the aim of the work is not very scientific. It fits more into a popular science study than a scientific article - it needs to be improved. Because at this stage it needs to be removed and changed.
Without a clear goal, we have no clear methods. Moreover, what is surprising is the lack of some necessary elements in the methods - why, for example, is there no information about to how authors' division of mosses into individual life forms - no citations? No detailed description of how it was done - there is nothing but results.
Another suggestion concerns Fig. 1 - is it really Fig. 1A, Fig. 1B, Fig. 1C - is this really what the reader is supposed to see, is it really that important, to put it on the drawing? I don't think so!
The results (partially) provide interesting data, and this is the best part of the manuscript... however, their method of presentation is quite surprising. An example are tables (Tables 1-5) - columnar tables are a waste of space and one and a half pages were used for such tables, where they could be described in a few sentences - combine tables or delete them.
Wasting space in the article and creating unnecessary Tables and Figures is common here and must be changed… such (another) example is Fig. 3 and Fig. 4, which present practically the same data… if only Fig. 4 is left, we have this the result itself and description in the manuscript. Another example is the incomprehensible to me the appearance of basic (at the bachelor's or master's level) errors in the presentation of results... lines 259-257 and Fig. 6 and Fig. 7 - it is known that, as you describe (in the text, all elements are not you already show them on such a detailed chart) - this definitely needs to be improved.
Another issue is the very superficial treatment of information about types? Is this supposed to be a new article that the authors are so stingy with information? Nomenclature types are one of the most important elements of herbaria, and here they are only a mention - I propose to expand on this topic.
The discussion begins with a paragraph that is more of a statement and does not contain any literature... it is difficult for me to continue to evaluate this manuscript which, in my opinion, has so many very basic errors. The subsequent discussion is weak and very short… probably too short!
The conclusions do not reflect the text and are very general comments that can be written about any herbarium... I see no point in trying to draw conclusions if there are none. Therefore, the conclusions should refer specifically to your manuscript, your herbarium and not herbaria as such...
To be honest, the number of basic mistakes is very large, and the topic, despite its potential, was wasted. Considering the number of basic errors, shortcomings, repetitions and the very, very small impact of this article on the field and the scientific community... I suggest you rethink the topic and write an article about it again.
I made more comments in the pdf file.

Reviewer 2 Report
Comments and Suggestions for Authors
The paper is fine in my opinion and could be published after correcting three small issues I detected:
1) The information in Table 1 (North America continent: 454 records) is inconsitent with the information in Table 2 (United States of America: 710) or the colors for Northamerican countries in Fig. 2 (dark blue for USA, ligth blue Canada, purple Mexico).
2) L265.- Change Dricanaceae to Dicranaceae
3) Figure 6.- Change Dricanaceae to Dicranaceae
Reviewer 3 Report
Comments and Suggestions for Authors
In the manuscript, the authors summarized the bryophytes collection at the UB Herbarium. The UB herbarium in Brazil holds 30, 902 specimens of bryophytes from 79 countries in all continents. The UB Herbarium plays a fundamental role in furture bryophytes taxonomy, systematics, biodiversity and so on.
Here are the problems I see:
1. This article should list there are how many families, genera, and species of mosses, liverworts or hornworts individually based on the present determined specimens? And the proportion of each category of bryophytes in the world? It is crucial for researchers and institutions interested in the study of bryophytes and biodiversity.
2. Since 1963, the UB Herbarium established, how many collectors contributed to the bryophytes collection? Because their collections enriched the bryophytes specimens of the herbarium. And the article should highlight their contributions.
3. Is there any studies used the bryophytes specimens of the UB herbarium for morphological measurements, DNA analyses etc. on previous work. If so, the authors should cite the related articles and analyze the relationship between their work and bryophytes specimens at the UB herbarium. And this will highlight the visibility and importance of the UB Herbarium.
4. Line 23, According to [2] ……. This cite method is not suitable. Knowledge about biodiversity allows us to understand the past and learn how to better manage the future [2].
5. Line 75 playing a crucial role in ecosystems {16].
Change the punctuation from {16] to [16].
I hope these comments are helpful and I wish you success in your continued research.
Comments on the Quality of English LanguageThe manuscript needs careful editing by someone with expertise in technical English editing paying particular attention to english grammar, spelling, and sentence structure so that the goals ans results of the study are clear to the reader.
Reviewer 4 Report
Comments and Suggestions for Authors
The presentation of bryophyte herbarium at UB is surely of interest to scientific community, however I believe that significant alterations of the text are necessary.
First, I believe that some parts of the text are not necessary at all (the long introductory passages on importance of herbaria, importance of bryophytes etc.). The non-relevant contents is perhaps also the paragraphs describing the distribution, ecology and threat to two species present in the collections of UB, and the Table 6, listing the 'rare and ancient collections' - it looks like a kind of random choice of specimens with older collection dates; similar specimens are plentiful in older established herbaria.
On the other hand, I was missing the information which I would find useful, such as the more detailed information on type specimens (collectors; whether holo/-lecto-neotypes are present among type materials etc.), a general information on collection dates (histogram) - this might be useful for the consideration of this herbarium in requests for material suitable for molecular analyses etc. It would be also useful to describe the handling protocols of the specimens and state of digitisation of records (I assume it is very high given the presented mapping results)
If you want to provide a listing of included families (in my opinion also not necessary), it is perhaps important to name the source of classification (as different authors would assign the genera to different families).
There are some obvious errors in the text (e.g., it is not possible that there are 14,081 South American specimens, Table 1, if Brazil only is represented by 19,553 records; misspellings in authors and wrong currently accepted generic assignations in Table 6).
Comments on the Quality of English LanguageAlthough not a native English speaker, I found the quality of English generally fine and adequate to be understood. It could still benefit from minor improvements.
Reviewer 5 Report
Comments and Suggestions for Authors
The manuscript is acceptable for publication after considering the comments below.
The abstract does not reflect your entire work. Please organise the abstract in a more conventional way: in addition to the introduction, highlight the objectives, methodology, results, discussion and conclusions.
The introduction is very rich in information, both of a general nature about the importance of herbaria and about the most important herbaria in the world, the description of the UB Herbarium of the University of Brasília and specifically the bryophyte collections housed in the UB Herbarium. Interesting, but maybe a little bit too long.
I would suggest a better description of the aims and objectives of the work.
For the Materials and Methods section, I would suggest the following:
- In the first two sentences it would be better to specify whether it is the Bryophytic Herbarium or the UB Herbarium in general, and in this respect the third sentence is not very clear: “The total number of specimens…………….
- The paragraph “The herbarium's abbreviations adhere to the Index Herbariorum” is out of place here. I would suggest moving it to the end of this section.
- Lines 146 and 147 the sentence: “Phytogeographic domains are based on [21].” This is incomplete, the bibliographic reference given is not sufficient.
Results:
In the legend to Figure 2, I think we should reverse the order of the values 299-100 to 100-200.
It is not clear to me whether the countries listed and the corresponding values in Table 2 correspond to those in Figure 2. It seems to me that this can only be deduced for the first two (Brazil and Papua New Guinea).
I think the comments from lines 304 to 310 are better suited to the discussion section.
Discussion:
I suggest that there should be better commentary on the results of map construction and data processing, and better justification of the objectives of the work.
Conclusions:
The conclusions are very general because they reflect the objectives of all herbaria and, in my opinion, they do not underline the aims of the authors. Therefore, I suggest improving the conclusions by also taking into account some considerations reported in the phytogeographical and conservation results of some rare or particularly interesting species, and by giving more emphasis to the possible validity of the proposed methodology in view of further necessary investigations.